# SARS-CoV-2 Breakthrough Infections after Third Doses Boost IgG Specific Salivary and Blood Antibodies

**DOI:** 10.3390/vaccines11030534

**Published:** 2023-02-24

**Authors:** María Noel Badano, Matias J. Pereson, Florencia Sabbione, Irene Keitelman, Natalia Aloisi, Roberto Chuit, María M. E. de Bracco, Susana Fink, Patricia Baré

**Affiliations:** 1Instituto de Medicina Experimental (IMEX)-Consejo Nacional de Investigaciones Científicas y Técnicas (CONICET), Academia Nacional de Medicina, Buenos Aires 1425, Argentina; 2Instituto de Investigaciones Hematológicas (IIHEMA), Academia Nacional de Medicina, Buenos Aires 1425, Argentina; 3Instituto de Investigaciones en Bacteriología y Virología Molecular (IBaViM), Facultad de Farmacia y Bioquímica, Universidad de Buenos Aires, Buenos Aires 1113, Argentina; 4Instituto de Investigaciones Epidemiológicas (IIE), Academia Nacional de Medicina, Buenos Aires 1425, Argentina

**Keywords:** SARS-CoV-2 breakthrough infections, vaccines, anti-spike antibody levels, humoral response, IgG specific salivary antibodies

## Abstract

SARS-CoV-2 breakthrough infections, associated with waning immunity, increase systemic antibody levels. In this study, we analyzed the impact of the infection timing on the magnitude of the systemic humoral response and whether breakthrough infections also boost antibody levels in the salivary compartment. We observed that the combination of infection plus vaccination, regardless of infection timing, produced a sharp increase in systemic antibodies, which were higher in subjects infected after third doses. Moreover, despite high systemic antibody levels, breakthrough infections after dose three occurred and boosted antibody levels in the salivary compartment. These results suggest that current vaccination strategies against COVID-19 should be improved. Results also showed that determination of salivary antibodies against SARS-CoV-2 could be a valuable tool in disease prevalence studies, for the follow-up of vaccinated individuals, and to assist vaccination strategies against COVID-19, especially in settings where blood sampling cannot be fulfilled.

## 1. Introduction

SARS-CoV-2 breakthrough infections in fully vaccinated individuals were associated with low systemic antibody levels, suggesting that third dose administration might reduce the risk of infection [1]. It was shown that one vaccine dose in previously infected individuals, as well as breakthrough infections in fully vaccinated subjects, increase systemic immunity [2,3,4,5]. However, information regarding systemic antibody levels in individuals infected between doses or after third doses is scarce.

The systemic humoral response against SARS-CoV-2 has been extensively studied, while less attention has been paid to the specific salivary antibodies. Some studies in SARS-CoV-2-infected subjects have shown a correlation between the systemic and salivary anti-SARS-CoV-2 IgG antibody levels [6,7], suggesting that salivary IgG could reflect the systemic antibody response. Studies have also reported specific salivary antibodies in individuals fully vaccinated mainly with COVID-19 mRNA vaccines, and they showed that specific salivary antibodies indicate seroconversion and correlate with serum neutralization [8,9]. Salivary antibodies represent mucosal responses that could be relevant in vaccine prevention of oral and nasal SARS-CoV-2 transmission.

In this work, we analyzed the impact of SARS-CoV-2 infection on the magnitude of the systemic humoral response in subjects infected before vaccination or after the first, second, or third vaccine doses. Salivary antibody levels before and following SARS-CoV-2 breakthrough infections were also investigated.

## 2. Materials and Methods

### 2.1. Study Design, Participants and Samples

Results presented in this report are part of an ongoing observational prospective cohort study started at the beginning of the pandemic among healthcare workers from the Academia Nacional de Medicina, in order to study the humoral immune response against SARS-CoV-2 after vaccination and/or infection.

Subjects included in this study had a confirmed diagnosis of SARS-CoV-2 infection at some point of vaccination. According to the moment of infection, they were classified into 4 groups (G): infected before vaccination (G0, *n* = 18), and after the first (G1, *n* = 6), second (G2, *n* = 10), and third vaccine dose (G3, *n* = 19). They all had mild COVID-19 based on the World Health Organization classification [10].

Blood samples were collected at different time points (T): before vaccination (T0) and 21–30 days after the first (T1), second (T2), and third (T3) doses. For subjects infected with SARS-CoV-2 after the first (G1), second (G2), or third vaccine doses (G3), blood samples were also obtained 21 to 30 days after infections (ai), referring to T1_ai_, T2_ai_, and T3_ai_, respectively. Saliva samples were collected at T3 and T3_ai_ from individuals infected after the third dose (G3, *n* = 17). Blood and saliva samples from uninfected control subjects vaccinated with three doses (CG, *n* = 18), matched for sex, age, and vaccine platform, were also obtained at T3 and 21–30 days after G3 infections (T3_c_).

Vaccine platforms and number of doses received according to each group are detailed in Table 1. Vaccines from the same platform were administered on similar dates and with similar time interval between doses (Appendix A).

Plasma or serum were obtained after centrifugation of the peripheral blood and stored in aliquots at −20 °C until used. For saliva sampling, individuals spat their first saliva of the day into a tube, without drinking, brushing teeth, or eating before collection. Saliva samples were centrifuged at 17,000× *g* for 10 min (4 °C) and the supernatant was stored at −20 °C until used.

This study was approved by the local Academia Nacional de Medicina Ethics Committee. Written informed consent was obtained from participants.

### 2.2. SARS-CoV-2 Antibody ELISA

SARS-CoV-2 spike-specific IgG antibodies were measured by ELISA (COVIDAR-IgG). The plates of the assay were coated with a purified mixture of the spike protein and the receptor binding domain (RBD) of the SARS-CoV-2 from the original viral variant from Wuhan (GenBank: MN908947). Antibody concentrations in binding antibody units (BAU) per mL (BAU/mL) were obtained, interpolating the OD 450 nm values of samples into a calibration curve constructed with the provided standard (400 BAU/mL). For serum samples, SARS-CoV-2 antibodies were determined following the manufacturer’s instructions (Laboratorios Lemos S.R.L, Buenos Aires, Argentina) [3]. As antibody levels are lower in saliva than in blood, the necessary conditions for salivary measurements were set by partially modifying those used for blood samples. Basically, salivary SARS-CoV-2 IgG antibodies were determined as plasma ones, without performing the first sample dilution. Since pre-pandemic samples were unavailable, negative controls comprised PCR-negative saliva samples from the early stages of the pandemic obtained before vaccination started in our country.

### 2.3. Statistical Analysis

Unpaired *t* tests or Mann–Whitney tests were used to compare the antibody levels between two groups. Categorical data were analyzed by the Chi-square test or Fisher’s exact test. The Spearman coefficient of rank correlation was used to assess the correlation between specific salivary and blood antibodies. For each time point, geometric mean concentrations (GMC) of specific antibody levels with 95% confidence intervals (95% CI) were calculated. A value of *p* < 0.05 was considered as a significant difference. Data analyses were performed using the GraphPad 9.1.2 Prism software (GraphPad Software, San Diego, CA, USA).

## 3. Results

Systemic antibody levels were analyzed in 4 groups of subjects infected with SARS-CoV-2 at different time points of vaccination (T): prior to vaccination (G0, *n* = 18), and after the first (G1, *n* = 6), second (G2, *n* = 10), and third vaccine dose (G3, *n* = 19).

The majority of the individuals infected before vaccination (G0) caught the disease by the time of Argentina’s first wave, when the variants of interest (VOI) or concern (VOC) were not yet circulating. All previously infected subjects seroconverted, showing a GMC of 159.5 BAU/mL (95% CI: 80.4–316.5) before vaccination (T0). Antibody levels within this group significantly increased after one vaccine shot (T1), showing an 11-fold rise (GMC: 1775 BAU/mL; 95% CI: 1101–2861; *p* < 0.0001) (Figure 1A).

The GMC at T1 in subjects infected after the first dose (G1) was 59.0 BAU/mL (95% CI: 18.2–191.5), although 3 participants showed no detectable antibodies (2 women vaccinated with BBIBP-CorV and 1 man vaccinated with Sputnik V). Most subjects within this group had the disease at the first peak of the Argentina’s second wave, when the circulating VOI/VOC alpha, gamma, and lambda predominated. Following infection (T1_ai_), a sharp increase in antibody concentrations was observed within this group, showing a GMC 64-fold higher than before infection (GMC: 3785 BAU/mL; 95% CI: 2014–7115; *p* = 0.005) (Figure 1A). 

SARS-CoV-2 infections after the second dose occurred mainly by the end of Argentina’s second wave, when the VOI/VOCs alpha, gamma, lambda, and delta were in circulation. The GMC at T2 in participants who subsequently became infected after the second dose (G2) was 167.5 BAU/mL (95% CI: 78.7–356.5) and increased 20-fold after infection (T2_ai_) (GMC: 3401 BAU/mL; 95% CI: 2135–5419; *p* < 0.0001) (Figure 1A).

Breakthrough infections after third doses occurred during Argentina’s third wave, when the delta variant was completely replaced by omicron [11]. This group (G3) showed high anti-SARS-CoV-2 antibody titers after the third dose (T3) (GMC: 1619 BAU/mL; 95% CI: 1085–2418), which increased even more following infections (T3_ai_), reaching a GMC 4.4-fold higher than before infection (GMC: 7099 BAU/mL; 95% CI: 5598–9001; *p* < 0.0001) (Figure 1A). These antibody levels were higher than those after one vaccine dose (T1) in subjects with prior SARS-CoV-2 infection (*p* < 0.0001) or following infections after dose 1 (T1_ai_, *p* = 0.04) or 2 (T2_ai_, *p* = 0.01) (Figure 1A).

Antibody concentrations in saliva were analyzed in cases (G3) and in uninfected control subjects vaccinated with three doses (CG), before (T3) and following breakthrough infections after third doses (T3_ai_ and T3_c_, respectively). At T3, IgG specific salivary antibodies were present in 82% of cases (GMC: 40.9 BAU/mL; 95% CI: 15.9–105.1) and in 74% of uninfected controls (GMC: 29.2 BAU/mL; 95% CI: 10.5–81.4). After breakthrough infections (T3_ai_), salivary antibodies were detected in 100% of cases, showing a 14-fold increase in the GMC (GMC: 570.7 BAU/mL; 95% CI: 369.0–882.7; *p* < 0.0001). These antibody titers were also higher than those observed at the same time point (T3_c_) in uninfected control subjects ([GMC: 21.7 BAU/mL; 95% CI: 10.0–80.4; (*p* < 0.0001)] (Figure 1B), remaining detectable in 72% of them (Figure 1B). A positive correlation between specific salivary and blood antibodies was observed (*r* = 0.61, *p* < 0.0001) (Figure 1C).

## 4. Discussion

The systemic antibody concentrations following vaccination in subjects with prior SARS-CoV-2 infection have been widely studied [2,3]. A strong boost of the systemic humoral response after breakthrough infections in fully vaccinated individuals has also been reported [4,5]. However, antibody concentrations in individuals infected with SARS-CoV-2 between doses or after third doses have been barely investigated.

In this study, we analyzed the impact of infection timing on the magnitude of the systemic humoral response. We observed a sharp increase in systemic antibodies following infections after the first, second, or third vaccine doses, reaching GMCs 64-, 20-, and 6.8-fold higher than before infection. These results showed that SARS-CoV-2 infection at any timepoint of vaccination boosts systemic antibody levels. Moreover, antibody levels following breakthrough infections after dose three were higher than in subjects vaccinated and infected at any previous time point. These results agree with a study showing that increased number of exposures to SARS-CoV-2 antigens, through vaccination or infection, enhance serum-specific, antibody-binding titers [4].

Antibody detection in saliva has been used to monitor exposure to pathogens, since the antibody profiles in saliva match well with those found in blood [12]. Although IgA is the key immunoglobulin for mucosal immunity, evidence for salivary IgA specific against SARS-CoV-2 is inconsistent. Salivary IgG is derived primarily from circulating IgG through transudation, whereas salivary IgA can be produced locally [13]. Therefore, IgG antibody levels in saliva could reflect the systemic antibody response, although at lower concentrations. In agreement, it has been observed that the 100% of individuals receiving full vaccination schedules with SARS-CoV-2 mRNA vaccines showed IgG-specific salivary antibodies, while only the 55% of subjects had IgA-specific salivary antibodies [8].

In this report, we analyzed salivary antibody levels before and after SARS-CoV-2 breakthrough infections in cases and in uninfected vaccinated control subjects. We observed that most subjects showed IgG-specific salivary antibodies at the time of infection, albeit at much lower levels than those in blood. Following infection, salivary antibody levels increased in cases, being significantly higher than before infection or in uninfected controls. These results show that SARS-CoV-2 breakthrough infections boost IgG antibody levels both in blood and saliva. Whether this boost of specific salivary antibodies could confer immune protection for future exposures remains to be determined. As previously reported [8,9], we observed that the levels of specific antibodies in saliva and blood correlated positively. Although not analyzed in this study, immune boosting by the infection in subjects vaccinated with whole virus COVID-19 vaccines might also increase systemic and salivary antibodies against other SARS-CoV-2 proteins. It is important to note that the increase in systemic and salivary antibody levels after infection was observed in subjects receiving different vaccines, suggesting that it was independent of vaccine platform. However, the small number of subjects precluded further statistical analysis to test potential associations between antibody levels achieved after infection and the vaccine platforms received.

Limitations of this study include the small number of subjects within groups, lack of information on neutralization titers, cellular immune responses, viral variants involved in the infections, and their associations with the antibody levels reached. Moreover, although salivary antibodies were not measured from the beginning of the study, our preliminary results show that four months after the third dose, salivary antibody concentrations in subjects infected at different time points of vaccination are higher than in uninfected vaccinated subjects (unpublished data).

While an enhanced neutralizing activity against the omicron variant was demonstrated after the administration of third doses [4], infections with this viral variant after third doses occurred [14], in agreement with our results. SARS-CoV-2 breakthrough infections were associated with waning immunity. However, in our cohort, infections after third doses happened despite high systemic antibody levels (GMC: 1619 BAU/mL; 95% CI: 1085–2418). These results lead to rethink current vaccination strategies in order to analyze how many doses and vaccine types are necessary to control viral infections by emerging variants, and to support the new vaccine candidates’ development targeting these variants.

Saliva collection is an attractive alternative to blood testing as it is non-invasive and allows home-based self-collection. This is important for studies in pediatric populations or where blood sampling is not possible. However, one limitation for the detection of specific salivary antibodies is that their levels are lower than in serum. In this report, we showed that salivary antibody titers increased after SARS-CoV-2 breakthrough infections. Therefore, determination of salivary antibodies against SARS-CoV-2 could be a valuable tool in disease prevalence studies, for the follow-up of vaccinated individuals, and to assist vaccination strategies against COVID-19, both in adult and pediatric populations, especially in settings where blood sampling cannot be fulfilled.

## 5. Conclusions

In summary, our results showed that the combination of SARS-CoV-2 infection plus vaccination, regardless of infection timing, produced a sharp increase in antibody levels, which was reflected both in the blood and the salivary compartment.

## Figures and Tables

**Figure 1 vaccines-11-00534-f001:**
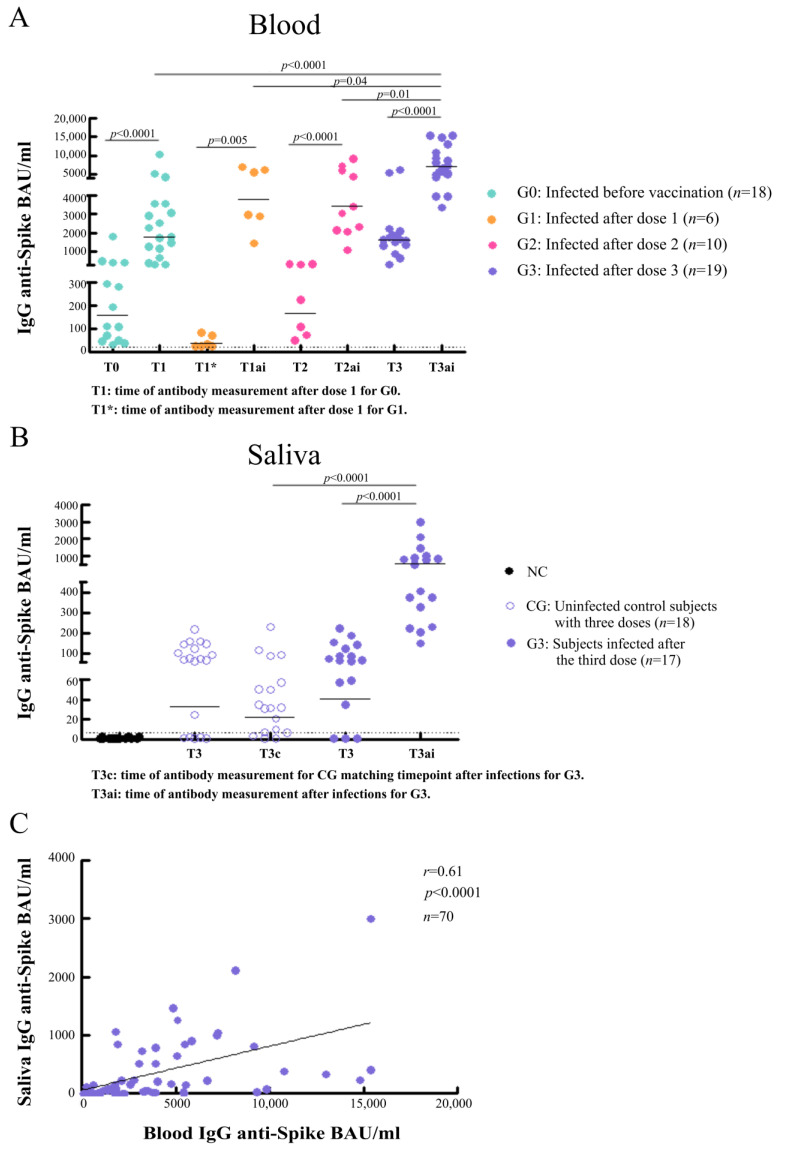
SARS-CoV-2 infection boosts humoral immunity in vaccinated individuals. (**A**) Systemic antibody concentrations in subjects infected with SARS-CoV-2 before vaccination (G0) or after the first (G1), second (G2), or third (G3) vaccine doses. SARS-CoV-2 spike-specific IgG antibodies were measured before vaccination (T0) and 21–30 days after the first (T1), second (T2), and third (T3) doses. For subjects infected with SARS-CoV-2 after the first (G1), second (G2), or third vaccine doses (G3), antibody concentrations were also measured 21 to 30 days after infections (ai), identified as T1_ai_, T2_ai_, and T3_ai_ respectively. IgG anti-spike antibody concentrations (BAU/mL) with geometric means are shown. Dotted line indicates the assay detection limit (4.03 BAU/mL). *p* values were determined by unpaired *t* test or Mann–Whitney test. (**B**) Salivary IgG antibodies to SARS-CoV-2 were measured in cases (G3) and in uninfected control subjects vaccinated with three doses (CG), before (T3) and following breakthrough infections after the third doses (T3_ai_ and T3_c_, respectively). PCR-negative saliva samples from the early stages of the pandemic were used as negative controls (NC) for reactivity. IgG anti-spike antibody concentrations (BAU/mL) with geometric means are shown. Dotted line indicates the assay detection limit (4.03 BAU/mL). *p* values were determined by Mann–Whitney test. (**C**) Correlation between matched salivary and blood SARS-CoV-2-specific IgG responses. Blood and saliva sample pairs (*n* = 70) from individuals infected after the third dose (G3, *n* = 17), and from uninfected control subjects vaccinated with three doses (CG, *n* = 18), were collected before (T3) and following breakthrough infections (T3_ai_ and T3_c_, respectively). Correlation between specific salivary and blood antibodies was analyzed by the Spearman rank correlation test. Spearman correlation coefficient (*r*) and *p*-value are indicated.

**Table 1 vaccines-11-00534-t001:** Characteristics of subjects infected with SARS-CoV-2 at different time points of vaccination.

	Infection before Vaccination (*n* = 18)	Infection after Dose 1 (*n* = 6)	Infection after Dose 2 (*n* = 10)	Infection after Dose 3 (*n* = 19)	Uninfected (*n* = 18)
**General characteristics**					
Age (years)	41 (26–63)	38 (27–49)	39 (28–52)	44 (27–53)	44 (28–60)
Sex, *n* (%) Female/Male	7/11 (39/61)	4/2 (67/33)	5/5 (50/50)	12/7 (63/37)	11/7 (61/39)
**SARS-CoV-2-related characteristics**					
Time from infection to dose 1 (months)	6 (1–12)	-	-	-	-
SARS-CoV-2 infection after vaccine (days)	-	43 (31–50)	58 (18–308)	57 (22–77)	-
**Vaccine platforms before/after infection**					-
**One dose**					-
BBIBP-CorV, *n* (%)	7 (39)	4 (66)	-	-	-
Sputnik V, *n* (%)	10 (56)	1 (17)	-	-	-
ChAdOx1 nCoV-19, *n* (%)	1 (5)	1 (17)	-	-	-
**Two doses**					-
BBIBP-CorV × 2, *n* (%)	-	-	6 (60)	-	-
Sputnik V × 2, *n* (%)	-	-	4 (40)	-	-
**Three doses**					
BBIBP-CorV × 2 + ChAdOx1 nCoV-19 × 1, *n* (%)	-	-	-	13 (68)	12 (66)
Sputnik V × 2 + ChAdOx1 nCoV-19 × 1, *n* (%)	-	-	-	3 (16)	3 (17)
ChAdOx1 nCoV-19 × 3, *n* (%)	-	-	-	3 (16)	3 (17)

Values are expressed as median (range) or *n* (%). BBIBP-CorV (Sinopharm); Sputnik V (Gamaleya NRCEM); ChAdOx1 nCoV-19 (University of Oxford/AstraZeneca).

## Data Availability

The datasets generated during and/or analyzed during the current study are available from the corresponding author on reasonable request.

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
