# Peer review of "SARS-CoV-2 Breakthrough Infections after Third Doses Boost IgG Specific Salivary and Blood Antibodies"

_vaccines, 2023, doi:10.3390/vaccines11030534_

Round 1

Reviewer 1 Report

  • Major comments: 

In the manuscript, M. Noel Badano et al. analyzed the impact of the infection timing on the magnitude of the systemic humoral response, and found that the combination of infection plus vaccination, regardless of infection timing, produced a sharp increase of systemic antibodies, especially in subjects infected after third doses, which also had boosted antibody levels in the salivary compartment.

The study is meaningful by revealing the increased systemic and salivary anti-spike antibodies in cases of SARS-CoV-2 breakthrough infections.

However, the scientifical and technical soundness of the study warrants further improvement.

The study can be further improved by including the following suggestions listed in specific comments.

  • Specific comments:

1)      The 53 subjects included in this study were vaccinated by different vaccines, it is questionable that whether the Anti-spike IgGs in these cases can be comparable.

2)      The 53 subjects included in this study may be reinfected by different SARS-CoV-2 variants. Thus, it is doubtful whether the increased Anti-spike IgG level triggered by different SARS-CoV-2 variants can be comparable.

3)      How many blood samples were drawn for the study?

4)       Line 79-80, why the B.1 variant Spike/RBD was used for the ELISA assay? The accession number should be specified. Besides, why the ELISA plates were coated with a mixture of the spike protein and the receptor binding domain (RBD) rather than a pure spike or RBD?

5)      Line 85-90, why the IgA level was not measured in saliva samples?

6)      Line 106-109 and Figure 1A, it is confusing here whether the increased antibody levels in T1 were caused by vaccination or infection?

7)      Line 116, the word “infected” should be omitted here according to the main text.

8)      In figure 1C, n=60? What samples were included in figure 1C?

9)      Line 142, please define the detection limit.

Reviewer 2 Report

The declared objective is to elucidate the impact of the infection timing on the magnitude of the systemic and local humoral response. In fact the study is analyzing the impact of infection on the magnitude of the humoral response elicited by different doses of SARS CoV-2 vaccines, which is a completely different thing, and show that infection after a 3rd dose is eliciting a lower increase in the antibody titer compared to infection after a 2nd dose. It is unclear if all subjects received 3 doses of vaccines, what was the dynamic of the antibody response after 2 and 3 vaccine doses, if there where differences according to the prior infection status. Are there any differences in terms of antibody persistence between individuals infected after the 2nd/3rd  dose  and those uninfected?  There is a big variability in the lengths of the interval between the 2nd dose of vaccine and infection which might influence the seroconversion level. Moreover, different types of vaccines have been used, different types of boosters, there are no comments on the potential differences in the efficacy of these vaccine types/ combinations.

Reviewer 3 Report

The authors studied whether the timing of infection with Sars Cov-2 (before vaccination, between doses or after complete vaccination) affects the production of specific antibodies in sera and in saliva. They observed that the combination of infection plus vaccination, regardless of infection timing, generates a sharp increase of systemic antibodies, being higher in subjects infected after three doses. Besides, breakthrough infections after dose 3 boost antibody levels in saliva. The results highlight the relevance of the assessment of salivary anti-SARS-CoV-2 antibodies for disease surveillance and vaccination follow-up, which is well pointed at the Discussion section. Moreover, the assessment of the immunological response in saliva is especially relevant due to the current search for prevention of SARS-Co-V-2 transmission.

The manuscript is easy to read, the Introduction is concise, but complete and very clear and the Methods and Results are well explained. The limitations of the study are well detailed, and all conclusions are strictly based on the results.

Minor:

I have found this part of Table 1 difficult to understand: SARS-CoV-2-related characteristics. Time since infection to dose 1 (months) it means from infection?

Please correct: Table 1 One doce

Discussion Line 166 Our results suggest that SARS-CoV-2 infection increases systemic antibody levels in vaccinated subjects, regardless of whether the exposure occurs before, during or after vaccination.” Based on the results shown, I understand this is shown, not suggested.

Author Response

Dear Reviewer,

We appreciate your comments and have improved our paper based on your suggestions. Revision has been made according to your suggestions:

1) I have found this part of Table 1 difficult to understand: SARS-CoV-2-related characteristics. Time since infection to dose 1 (months) it means from infection?

Authors’ response:

Taking into consideration your observation, the word “since” was replaced by “from” in the table.

2) Please correct: Table 1 One doce

Authors’ response:

Considering your remark, this correction was made.

3) Discussion Line 166“Our results suggest that SARS-CoV-2 infection increases systemic antibody levels in vaccinated subjects, regardless of whether the exposure occurs before, during or after vaccination.” Based on the results shown, I understand this is shown, not suggested.

Authors’ response:

As suggested by the reviewer, the phrase was changed to (Line 179): “Our results showed that SARS-CoV-2 infection increases systemic antibody levels in vaccinated subjects, regardless of whether the exposure occurs before, during or after vaccination.

Reviewer 4 Report

The manuscript by Badano et al., an interesting study and can provide a way to observe the antibody level of blood and saliva after SARS-CoV-2 infection plus vaccination. And whether the antibody produced can play a protective role against SARS-CoV-2 infection. The main problem of the full text is that the number of samples is small (but it can meet the needs of statistical calculation). Based on the type of the paper is brief report, this shortcoming is acceptable. Therefore, the full text meets the requirements of publication.

Author Response

Dear reviewer,
We appreciate your comments and the time taken to review our manuscript.

Reviewer 5 Report

General recommendations:

Analysis of the impact of the SARS-CoV-2 infection on the magnitude of the systemic humoral immune response and the comparison with the magnitude of response observed in the salivary compartment is an interesting concept. However, I'm a bit confused by the group selection and classification in this study and the manuscript is difficult to follow at first, especially in the methodology and results section. Also, it seems to me that the authors didn't explain quite well the timepoint when they took the samples, because at the end of the discussion, some four months are mentioned. Furthermore, I’ve checked the manual of the test used in this study, and I believe that this test is only validated for serum and plasma, but not for saliva, which is potentially problematic.

Major comments:

1.       Materials and Methods section, in particular the study design, selection of participants (inclusion and exclusion criteria) and sampling (especially timepoints) should be re-written in a way to improve understanding of this very important section (L53-L71).

2.       L81-83: Is COVIDAR-80 IgG ELISA validated for saliva samples?

3.       Results section is also difficult to follow, for example:

a)       L112-115: What prior study was referred to? There was no reference (and even if there was, statement such as this usually belongs in the Discussion section)

b)      L116-122: It is difficult to follow what is before and what after the infection. The entire paragraph needs to be re-written to improve understanding. Maybe it would be better to say the value of the GMC after the 1st dose, and then to say that the most of the participants was infected at XX timepoint (in relation to the vaccine dosage).

4.       Figure 1B) : The labelling is unclear, why would the authors label the uninfected group as T3ai during the follow-up? They did not have an infection, they were just taken as a re-control for G3, the group that actually had breakthrough infections.

5.       Figure 1C) : It would be perhaps more effective to evaluate the correlation before and after the infection.

Minor comments:

1.       L25-27: Please rephrase the entire sentence because it doesn’t make sense the way it is written now.

2.       L97: What kind of test is used to determine the type of data distribution?

3.       L183-185: Please rephrase the entire sentence because it doesn’t make sense the way it is written now.

4.       L211: There is no need for the quotation marks.

Author Response

Dear Reviewer,

We appreciate your comments and have improved our manuscript accordingly. Revision has been made based on your suggestions:

1) It seems to me that the authors didn't explain quite well the timepoint when they took the samples, because at the end of the discussion, some four months are mentioned.

Authors’ response:

Samples were collected 21-30 days after each dose and 21-30 days after infection.

The four months mentioned in discussion section (Line 293) refer to unpublished data and not to the results presented herein. Therefore, this information was added to the original sentence of the manuscript (Line 295): “Besides, although salivary antibodies were not measured from the beginning of the study, our preliminary results show that four months after the third dose, salivary antibody concentrations in subjects infected at different time points of vaccination, are higher than in uninfected vaccinated subjects (unpublished data)”.

2) Materials and Methods section, in particular the study design, selection of participants (inclusion and exclusion criteria) and sampling (especially timepoints) should be re-written in a way to improve understanding of this very important section (L53-L71).

Authors’ response:

The content of the subtitle Study design, participants and samples in Materials and methods section was changed to (Line 78):

Study design, participants and samples

Results presented in this report are part of an ongoing observational prospective cohort study started at the beginning of the pandemic among healthcare workers from the Academia Nacional de Medicina, to study the humoral immune response against SARS-CoV-2 after vaccination and/or infection.

Subjects included in this study had a confirmed diagnosis of SARS-CoV-2 infection at some point of vaccination. According to the moment of infection, they were classified into 4 groups (G): infected before vaccination (G0, n=18), after the first (G1, n=6), second (G2, n=10) and third vaccine dose (G3, n=19). They all had mild COVID-19 based on the World Health Organization classification [10]

Blood samples were collected at different time points (T): before vaccination (T0) and 21-30 days after the first (T1), second (T2) and third (T3) doses. For subjects infected with SARS-CoV-2 after the first (G1), second (G2) or third vaccine doses (G3), blood samples were also obtained 21 to 30 days after infections (ai), referring to T1ai, T2ai and T3ai respectively. Saliva samples were collected at T3 and T3ai from individuals infected after the third dose (G3, n=17). Blood and saliva samples from uninfected control subjects vaccinated with three doses (GC, n=18), matched for sex, age and vaccine platform, were also obtained at T3 and 21-30 days after G3 infections (T3c).

Vaccine platforms and number of doses received according to each group are detailed in Table 1. Vaccines from the same platform were administered on similar dates and with similar time interval between doses”.

3) L81-83: Is COVIDAR-80 IgG ELISA validated for saliva samples?

Authors’ response:

COVIDAR-IgG is validated for serum and plasma. However, some publications demonstrate that salivary antibodies can be detected by commercial anti-SARS-CoV-2 serology tests by making minimal modifications consisting mainly of using a lower sample dilution, as antibody concentrations in saliva are lower than in blood (Azzi et. al., doi: 10.1016/j.ebiom.2021.103788; Terreri et. al, doi: 10.1016/j.chom.2022.01.003; Sundar et. al., doi: 10.3390/vaccines10111819; Mancuso et.al, doi: 10.3390/vaccines10101649). It is important to mention that these assays were also used to measure specific antibodies in nasal swabs (Collier et. al, doi: 10.1126/scitranslmed.abn6150) and breastmilk samples, as performed by the developer of the ELISA test COVIDAR-IgG (Longueira et. al, doi: 10.3389/fimmu.2022.909995).

4) L112-115: What prior study was referred to? There was no reference (and even if there was, statement such as this usually belongs in the Discussion section)

Authors’ response:

To clarify that, the phrase was changed to (Line 141): “Antibody levels within this group significantly increased after one vaccine shot (T1), showing a 11-fold rise (GMC: 1775 BAU/mL; 95% CI: 1101-2861; p<0.0001) (Figure 1A)”.

5) L116-122: It is difficult to follow what is before and what after the infection. The entire paragraph needs to be re-written to improve understanding. Maybe it would be better to say the value of the GMC after the 1st dose, and then to say that the most of the participants was infected at XX timepoint (in relation to the vaccine dosage)

Authors’ response:

Thank you for your suggestion. We have already corrected the paragraph in the manuscript (Line 150): “The GMC at T1 in subjects infected after the first dose (G1) was 59.0 BAU/mL (95% CI: 18.2-191.5), although three participants showed no detectable antibodies (2 women vaccinated with BBIBP-CorV and 1 man vaccinated with Sputnik V). Most subjects within this group had the disease at the first peak of the Argentina’s second wave, when the circulating VOI/VOC alpha, gamma and lambda predominated. Following infection (T1ai), a sharp increase in antibody concentrations was observed within this group, showing a GMC 64-fold higher than before infection (GMC: 3785 BAU/mL; 95% CI: 2014-7115; p=0.005) (Figure 1A)”.

6) Figure 1B) : The labelling is unclear, why would the authors label the uninfected group as T3ai during the follow-up? They did not have an infection, they were just taken as a re-control for G3, the group that actually had breakthrough infections.

Authors’ response:

T3 and T3ai refer to the time when the antibody concentrations were measured, both for the control group and the G3 group.

To clarify that, the matching sample collection time for GC (21-30 days after G3 infections) was renamed T3c. These modifications were applied throughout the manuscript, including the figure and figure legend, to incorporate this term:

- The original paragraph in Results section was changed to (Line 178): ”Antibody concentrations in saliva were analyzed in cases (G3) and in uninfected control subjects vaccinated with three doses (GC), before (T3) and following breakthrough infections after third doses (T3ai and T3c, respectively). At T3, IgG specific salivary antibodies were present in 82% of cases (GMC: 40.9 BAU/mL; 95% CI: 15.9-105.1) and in 74% of uninfected controls (GMC: 29.2 BAU/mL; 95% CI: 10.5-81.4). After breakthrough infections (T3ai), salivary antibodies were detected in 100% of cases, showing a 14-fold increase in the GMC (GMC: 570.7 BAU/mL; 95% CI: 369.0-882.7; p<0.0001). These antibody titers were also higher than those observed at the same time point (T3c) in uninfected control subjects (GMC: 21.7 BAU/mL; 95% CI: 10.0-80.4; (p<0.0001) (Figure 1B), remaining detectable in 72% of them (Figure 1B)”.

- Figure legend was changed to: “(A) Systemic antibody concentrations in subjects infected with SARS-CoV-2 before vaccination (G0) or after the first (G1), second (G2) or third (G3) vaccine doses. SARS-CoV-2 spike-specific IgG antibodies were measured before vaccination (T0) and 21-30 days after the first (T1), second (T2) and third (T3) doses. For subjects infected with SARS-CoV-2 after the first (G1), second (G2) or third vaccine doses (G3), antibody concentrations were also measured 21 to 30 days after infections (ai), identified as T1ai, T2ai and T3ai respectively. IgG anti-spike antibody concentrations (BAU/mL) with geometric means are shown. Dotted line indicates the assay detection limit (4.03 BAU/mL). p values were determined by Unpaired t test or Mann-Whitney test. (B) Salivary IgG antibodies to SARS-CoV-2 were measured in cases (G3) and in uninfected control subjects vaccinated with three doses (GC), before (T3) and following breakthrough infections after third doses (T3ai and T3c, respectively). PCR-negative saliva samples from the early stages of the pandemic were used as negative controls (NC) for reactivity. IgG anti-spike antibody concentrations (BAU/mL) with geometric means are shown. Dotted line indicates the assay detection limit (4.03 BAU/mL). p values were determined by Mann-Whitney test. (C) Correlation between matched salivary and blood SARS-CoV-2-specific IgG responses. Blood and saliva sample pairs (n=70) from individuals infected after the third dose (G3, n=17) and from uninfected control subjects vaccinated with three doses (GC, n=18), were collected before (T3) and following breakthrough infections (T3ai and T3c, respectively). Correlation between specific salivary and blood antibodies was analyzed by the Spearman rank correlation test. Spearman correlation coefficient (r) and p-value are indicated”.

7) Figure 1C) : It would be perhaps more effective to evaluate the correlation before and after the infection.

Authors’ response:

Correlation using only data from the G3 group (N=34), results in a coefficient r= 0.5 and p=0.004. Therefore, correlation was performed with blood and saliva sample pairs (n=70) from individuals infected after the third dose (G3, n=17) and from uninfected control subjects with three doses (GC, n=18), collected before (T3) and following breakthrough infections (T3ai and T3c, respectively), which results in a coefficient r= 0.61 and p<0.0001.

8) L25-27: Please rephrase the entire sentence because it doesn’t make sense the way it is written now.

Authors’ response:

Regarding this appreciation, the sentence in the manuscript was changed to (Lines 29-33): “These results suggest that current vaccination strategies against COVID-19 should be improved. Results also showed that determination of salivary antibodies against SARS-CoV-2 could be a valuable tool in disease prevalence studies, for the follow-up of vaccinated individuals and to assist vaccination strategies against COVID-19, especially in settings where blood sampling cannot be fulfilled”.

9) L97: What kind of test is used to determine the type of data distribution?

Authors’ response:

We used the D'Agostino-Pearson omnibus normality test provided by the GraphPad 9.1.2 Prism software.

10) L183-185: Please rephrase the entire sentence because it doesn’t make sense the way it is written now.

Authors’ response:

Authors’ response:

Regarding this appreciation, the paragraph in the manuscript was changed to (Lines 254-262): “In this study, we analyzed the impact of infection timing on the magnitude of the systemic humoral response. We observed a sharp increase of systemic antibodies following infections after the first, second or third vaccine doses, reaching GMCs 64-, 20- and 6.8-fold higher than before infection. These results showed that SARS-CoV-2 infection at any timepoint of vaccination boosts systemic antibody levels. Moreover, antibody levels following breakthrough infections after dose 3 were higher than in subjects vaccinated and infected at any previous time point. These results agree with a study showing that increased number of exposures to SARS-CoV-2 antigens, through vaccination or infection, enhance serum specific antibody-binding titers [4]”.

11) L211: There is no need for the quotation marks.

Authors’ response:

This was corrected.

Round 2

Reviewer 1 Report

The authors have addressed some of my concerns. However, some problems still need to be addressed:

1. Line 112-116, and Figure 1A, it seems like the GMC at T1(subjects infected after the first dose) was so low (59.0 BAU/mL), even lower than the T0 group, which had a GMC of 159.5 BAU/mL (95% CI: 80.4-316.5) before vaccination?

2. It seems like T0, T1, T2, T3, and T11/2, T11/2, T11/2  refer to time here? It would not be very clear sometimes in the study. Suggest using other symbols to make the description more clear.

3. The time of vaccination and infection of each individual should be disclosed.

4. Since the individuals in the study received different kinds of vaccines and they might be infected by different SARS-CoV-2 variants, it would be difficult to directly compare the IgGs results.

Author Response

Dear Reviewer,

We appreciate your comments and have improved our paper based on your suggestions. Revision has been made according to your suggestions:

1) Line 112-116, and Figure 1A, it seems like the GMC at T1(subjects infected after the first dose) was so low (59.0 BAU/mL), even lower than the T0 group, which had a GMC of 159.5 BAU/mL (95% CI: 80.4-316.5) before vaccination?

2) It seems like T0, T1, T2, T3, and T11/2, T11/2,T11/2  refer to time here? It would not be very clear sometimes in the study. Suggest using other symbols to make the description more clear.

Authors’ response:

To clarify that, we combined your first two questions in one response. As you said, the “Ts” are referring to time in the study and not groups.

In this work we analyzed the impact of SARS-CoV-2 infection on the magnitude of the systemic humoral response in 4 groups of subjects infected with SARS-CoV-2 at different time points: prior to vaccination (n=18), after the first (n=6), second (n=10) or third vaccine dose (n=19).

For this purpose, blood samples were drawn before vaccination (T0) and 21-30 days after the first (T1), second (T2) and third (T3) doses. For subjects infected with SARS-CoV-2 after the first, second or third doses, blood samples were also drawn 21 to 30 days after infection (ai: T1ai, T2ai and T3ai respectively).

Systemic antibody levels were analyzed at different time points of vaccination (T) for each group:

- for subjects infected before vaccination, antibody levels were analyzed at times T0 and T1.

- for individuals infected after the first dose, antibody levels were analyzed at times T1 and T1ai.

- for subjects infected after the second dose, antibody levels were analyzed at times T2 and T2ai.

- for individuals infected after the third dose, antibody levels were analyzed at times T3 and T3 ai.

Therefore, the following corrections were made to the manuscript:

-Groups were redefined as: G0: subjects infected before vaccination, G1: individuals infected after the first dose, G2: subjects infected after the second dose and G3: individuals infected after the third dose. These modifications were applied throughout the entire manuscript including the table, figure and figure legend.

- The subtitle Study design, participants and sample collection in Materials and methods section (Line 52) was changed to:

Study design, participants and samples

Since the beginning of the pandemic our laboratory collected blood and saliva samples from healthcare workers belonging to the Academia Nacional de Medicina, to detect and follow-up those subjects who became infected with SARS-CoV-2 and to study the immune humoral response over time after vaccination and/or infection. Therefore, in this study we included 4 groups of subjects infected with SARS-CoV-2 at different time points (T) of vaccination: before vaccination (G0, n=18), after the first (G1, n=6), second (G2, n=10) and third vaccine dose (G3, n=19). All had mild COVID-19 based on the World Health Organization classification [10]. The vaccine platforms and the number of doses received according to each group are detailed in Table 1. The same vaccine platforms were administered on similar dates and with similar interval time between doses.

Blood samples used in this study were those obtained before vaccination (T0) and 21-30 days after the first (T1), second (T2) and third (T3) doses. For subjects infected with SARS-CoV-2 after the first (G1), second (G2) or third vaccine doses (G3), blood samples collected 21 to 30 days after infections (ai: T1ai, T2ai and T3ai respectively) were also used.

Saliva samples were those collected at T3 and T3ai from individuals infected after the third dose (G3, n=17). Blood and saliva samples from uninfected subjects (GC, n=18), matched for sex, age, vaccine platform and collection time (T3 and T3ai), were used as controls”.

- The original sentence in Results section (Line 123) was changed to: “Systemic antibody levels were analyzed in 4 groups of subjects infected with SARS-CoV-2 at different time points of vaccination (T): prior to vaccination (G0, n=18), after the first (G1, n=6), second (G2, n=10) and third vaccine dose (G3, n=19)”.

- The original sentence in Results section (Line 168) was changed to: “Antibody concentrations in saliva were analyzed at T3 and T3ai in cases (G3) and in uninfected vaccinated control subjects (GC). At T3, IgG specific salivary antibodies were present in 82% of cases (GMC: 40.9 BAU/mL; 95% CI: 15.9-105.1) and in 74% of uninfected controls (GMC: 29.2 BAU/mL; 95% CI: 10.5-81.4). At T3ai, salivary antibodies were detected in 100% of cases, showing a 14-fold increase in the GMC (GMC: 570.7 BAU/mL; 95% CI: 369.0-882.7; p<0.0001), while remaining detectable in 72% of controls (GMC: 21.7 BAU/mL; 95% CI: 10.0-80.4) (Figure 1B). Salivary antibody titers at T3ai were also higher in cases (p<0.0001) than in uninfected control subjects (Figure 1B)”.

- Figure legend was changed to: “(A) Antibody concentrations in the blood of subjects infected with SARS-CoV-2 before vaccination (G0) or after the first (G1), second (G2) or third (G3) vaccine doses. SARS-CoV-2 spike-specific IgG antibodies were measured before vaccination (T0) and 21-30 days after the first (T1), second (T2) and third (T3) doses. For subjects infected with SARS-CoV-2 after the first (G1), second (G2) or third vaccine doses (G3), antibody concentrations were also measured 21 to 30 days after infections (ai: T1ai, T2ai and T3ai respectively). IgG anti-spike antibody concentrations (BAU/mL) with geometric means are shown. Dotted line indicates the assay detection limit (4.03 BAU/mL). p values were determined by Unpaired t test or Mann-Whitney test. (B) Salivary IgG antibodies to SARS-CoV-2 were measured at T3 and T3ai in subjects with breakthrough infections after dose 3 (G3) and in uninfected control subjects vaccinated with three doses (GC). PCR-negative saliva samples from the early stages of the pandemic were used as negative controls (NC) for reactivity. IgG anti-spike antibody concentrations (BAU/mL) with geometric means are shown. Dotted line indicates the assay detection limit (4.03 BAU/mL). p values were determined by Mann-Whitney test. (C) Correlation between matched salivary and blood SARS-CoV-2-specific IgG responses. Blood and saliva sample pairs (n=70) from individuals infected after the third dose (G3, n=17) and from uninfected control subjects vaccinated with three doses (GC, n=18) were collected at T3 and T3ai. Correlation between specific salivary and blood antibodies was analyzed by the Spearman rank correlation test. Spearman correlation coefficient (r) and p-value are indicated”.

- A graphical abstract was included.

3) The time of vaccination and infection of each individual should be disclosed

Thank you for your suggestion. To clarify this point, we did the following:

a- A table with the dates of vaccination and infection of every subject was added for editorial consideration as Supplementary information.

b- An overview of these data is represented in the graphical abstract.

c- Information about infection timing for each group was added to the Results section:

-Line 128: The majority of the individuals infected before vaccination (G0) got the disease by the time of Argentina’s first wave, when the variants of interest (VOI) or concern (VOC) were not yet circulating.

-Line 135: Most subjects infected after the first dose (G1) had the disease at the first peak of the Argentina’s second wave, when the circulating VOI/VOC alpha, gamma and lambda predominated. The GMC at T1 in this group was 59.0 BAU/mL (95% CI: 18.2-191.5), although three participants showed no detectable antibodies (2 women vaccinated with BBIBP-CorV and 1 man vaccinated with Sputnik V). A sharp increase in antibody concentrations was observed following infection (T1ai), showing a GMC 64-fold higher than before infection (GMC: 3785 BAU/mL; 95% CI: 2014-7115; p=0.005) (Figure 1A).

- Line 147: SARS-CoV-2 infections after the second dose occurred mainly by the end of Argentina’s second wave when the VOI/VOC alpha, gamma, lambda and Delta were in circulation. The GMC at T2 in participants who subsequently became infected after the second dose (G2) was 167.5 BAU/mL (95% CI: 78.7-356.5) and increased 20-fold after infection (T2ai) (GMC: 3401 BAU/mL; 95% CI: 2135-5419; p<0.0001) (Figure 1A).

- Line 155: Breakthrough infections after third doses occurred during Argentina’s third wave, when the Delta variant was completely replaced by Omicron [11]. This group (G3) showed high anti-SARS-CoV-2 antibody titers after the third dose (T3) (GMC: 1619 BAU/mL; 95% CI: 1085-2418) which increased even more following infections (T3ai) reaching a GMC 4.4-fold higher than before infection (GMC: 7099 BAU/mL; 95% CI: 5598-9001; p<0.0001) (Figure 1A). These antibody levels were higher than those after one vaccine dose (T1) in subjects with prior SARS-CoV-2 infection (p<0.0001) or following infections after dose 1 (T1ai, p=0.04) or 2 (T2ai, p=0.01) (Figure 1A).

4) Since the individuals in the study received different kinds of vaccines and they might be infected by different SARS-CoV-2 variants, it would be difficult to directly compare the IgGs results.

Authors’ response:

Thank you for your comment. As you stated, different vaccine platforms lead to different antibody levels (Dashdorj et al., 2021, Cell Host & Microbe 29, 1738–1743). Besides, SARS-CoV-2 variants may also impact on antibody levels reached after infections. As reported in a recent work (BÅ‚aszczuk A et al, 2022), a significantly lower antibody titer was observed in patients with the Omicron variant infection compared to the Delta variant.

However, in this study differences were analyzed within the same group, before and after infection/vaccination. With the exception of the G0, which was more heterogeneous in vaccine platforms, the rest of the individuals in G1, G2 and G3 had received mainly a primary BBIBP-CorV scheme. Moreover, most SARS-CoV-2 infections within the same group happened in a limited period of time, suggesting that viral variants involved are expected to be the same. 

However, it is worth noting that when we compared antibody levels reached after infections between G3 and the other groups (Line 165), we cannot rule out that viral variants involved in the infections, as well as the number of exposures to SARS-CoV-2 antigens, could have an impact on the results.

Reviewer 2 Report

There are serious flaws in the interpretation of the results, which do not take into consideration the particularities of the immune response elicited after multiple vaccination doses.

Author Response

Dear Reviewer,

We appreciate your comments and have included some changes in the manuscript that will help to clarify our points:

1) There are serious flaws in the interpretation of the results, which do not take into consideration the particularities of the immune response elicited after multiple vaccination doses.

Regarding this appreciation, we are aware that multiple vaccination doses enhance the magnitude of the immune humoral response. However, in our work we observed that the combination of infection plus vaccination, regardless the moment of infection, leads to a sharp increase of systemic antibody levels. Neither vaccination nor infection alone achieve the antibody levels seen in vaccinated and infected subjects. In fact, our cohort of subjects received mainly a primary BBIBP-CorV scheme, which is less immunogenic compared to others vaccination schedules (Dashdorj et al., 2021, Cell Host & Microbe 29, 1738–1743). Despite this, individuals who received one or two doses of the BBIBP-CorV vaccine and who subsequently became infected, developed higher antibody titers than those who did not become infected and received three vaccine doses (Figure 1A, T1ai or T2ai vs T3).

A graphical abstract was included to show the groups of infected subjects, the time points at which they were studied, and the antibody levels achieved in each group.

Round 3

Reviewer 1 Report

Please upload the graphical abstract, as it was not shown in the updated files.

Reviewer 2 Report

I maintain my previous opinion.

Author Response

Dear reviewer. We appreciate the time taken to review our manuscript.